# Role of Metallic Adlayer in Limiting Ge Incorporation into GaN

**DOI:** 10.3390/ma15175929

**Published:** 2022-08-27

**Authors:** Henryk Turski, Pawel Wolny, Mikolaj Chlipala, Marta Sawicka, Anna Reszka, Pawel Kempisty, Leszek Konczewicz, Grzegorz Muziol, Marcin Siekacz, Czeslaw Skierbiszewski

**Affiliations:** 1Institute of High Pressure Physics, Polish Academy of Sciences, 01-142 Warsaw, Poland; 2Department of Electrical and Computer Engineering Cornell University, Ithaca, NY 14853, USA; 3Institute of Physics, Polish Academy of Sciences, 02-668 Warsaw, Poland

**Keywords:** doping, molecular beam epitaxy, surfactant, nitrides

## Abstract

Atomically thin metal adlayers are used as surfactants in semiconductor crystal growth. The role of the adlayer in the incorporation of dopants in GaN is completely unexplored, probably because *n*-type doping of GaN with Si is relatively straightforward and can be scaled up with available Si atomic flux in a wide range of dopant concentrations. However, a surprisingly different behavior of the Ge dopant is observed, and the presence of atomically thin gallium or an indium layer dramatically affects Ge incorporation, hindering the fabrication of GaN:Ge structures with abrupt doping profiles. Here, we show an experimental study presenting a striking improvement in sharpness of the Ge doping profile obtained for indium as compared to the gallium surfactant layer during GaN-plasma-assisted molecular beam epitaxy. We show that the atomically thin indium surfactant layer promotes the incorporation of Ge in contrast to the gallium surfactant layer, which promotes segregation of Ge to the surface and Ge crystallite formation. Understanding the role of the surfactant is essential to control GaN doping and to obtain extremely high *n*-type doped III-nitride layers using Ge, because doping levels >10^20^ cm^−3^ are not easily available with Si.

## 1. Introduction

One of the most important features of semiconductors is the possibility of changing its conductivity via doping. Since the beginning of research devoted to III-nitride structures, scientists have struggled with efficient *p*-type doping [1]. The main challenge was the hydrogen passivating magnesium dopant in layers grown under an ammonia atmosphere, like for metalorganic vapor phase epitaxy (MOVPE). The issue with *p*-type doping in GaN was solved by either electron irradiation [2,3] or post-growth annealing in a N_2_ atmosphere [4]. As an alternative to using ammonia as a source of nitrogen, nitrogen plasma sources can be used in molecular beam epitaxy (MBE). For this technique, relatively lower substrate temperatures and metal (group III) excess conditions are used [5,6], and no *p*-type activation is needed.

*n*-type doping, on the other hand, is straightforward in III-nitrides and realized using silicon (Si). As shown for MOVPE, silicon is an effective dopant up to mediocre concentrations but can lead to surface roughening close to compositions of $1.9\;\times \;10^{19}\;\frac{\mathrm{atom}}{{\mathrm{cm}}^{3}}$ [7,8]. An attractive alternative for *n*-type dopants in nitrides is germanium (Ge) [9,10,11,12]. This element can be used in a much wider doping spectrum without degrading surface morphology [7]. It was also reported that Ge doping leads to lower compensation than Si, resulting in higher carrier concentrations at high doping levels [13,14]. Ge doping was also used to significantly lower the refractive index of GaN [15]. Recently, GaN:Ge layers with Ge concentrations reaching $6.7\;\times \;10^{20}\;\frac{\mathrm{atom}}{{\mathrm{cm}}^{3}}\;$ were grown by plasma-assisted MBE [16] and $2.2\;\times \;10^{20}\;\frac{\mathrm{atom}}{{\mathrm{cm}}^{3}}\;$ by ammonia-MBE [17]. In Ga(Al)N:Ge tunnel junctions, Ge concentrations reaching $5.5\;\times \;10^{20}\;\frac{\mathrm{atom}}{{\mathrm{cm}}^{3}}$ [18] and MOVPE growth of GaN:Ge layers for TJ-LEDs reaching $2\;\times \;\frac{\mathrm{atom}}{{\mathrm{cm}}^{3}}$ [19] were reported. These results prove the effectiveness of the use of Ge as a high-concentration *n*-type dopant in nitrides.

Numerical calculations targeting differences between Ge and Si *n*-type dopants have concentrated on the formation energies and the self-compensation mechanisms for these dopants under metal- and nitrogen-rich conditions [20,21,22]. No pronounced difference between Ge and Si was found, indicating low self-compensation and high possible electron concentrations for both donors in GaN. Different consequences of dopant incorporation are considered by Baker et al. [14], where the advantage of Ge is identified as a larger atom size that introduces lower strain, leading to lower metal vacancy density, which is responsible for lower compensation.

An important and less explored topic is the mechanism that limits Ge concentration in III-nitride structures. For example, at about 700 °C, germanium is expected to have a high solubility of above 45% in pure indium and 55% in pure gallium, which is almost one order of magnitude higher than silicon in either metal at the same temperature [23]. This means that the solubility in liquid metal on top of the crystal should not be a limiting factor for the final doping concentration. In spite of that, early efforts to obtain GaN:Ge by plasma-assisted MBE reported the formation of both Ge and Ge_3_N_4_ precipitates for concentrations exceeding 4 × 10^20^ atoms/cm^3^ as reported by Hageman et al. [24]. More recently, the doping limit for plasma-assisted MBE of AlGaN:Ge was identified by Bougerol et al. [25]. The authors claim to have found the pronounced solubility limit for germanium in AlGaN layers with Al contents higher than 0.2, leading to the formation of Ge crystallites and Ge inclusions, mainly formed near defects or metal droplets on the surface. The abovementioned findings suggest that the limiting factor for Ge incorporation is solubility in crystal rather than the impact of surfactant or growth kinetics. Surprisingly, similar precipitates were also reported for metal-modulated epitaxy of GaN:Ge layers grown at a high growth rate, but the authors assumed the precipitates consisted of Ga rather than Ge [26].

In spite of a few works investigating the possibility of achieving high doping concentrations of Ge in GaN cited above, there are no reports on the use of Ge-doped layers in p-n diodes, where such layers need to be overgrown with *p*-type layers.

In this work, we studied the incorporation mechanism of Ge into III-nitride layers during the plasma-assisted MBE process. We compared doping profiles obtained for GaN:Ge and low-In-content InGaN:Ge heterostructures and found higher dopant incorporation and sharper doping transient for InGaN. During GaN growth, a considerable number of germanium atoms tended to stay at the surface rather than being incorporated. The use of indium (instead of gallium) as a surfactant during growth resulted in no Ge-related precipitates [25]. By growing low-indium-content InGaN:Ge, we present one of the highest electron concentrations obtained in nitrides.

## 2. Materials and Methods

All samples in this study were grown by plasma-assisted MBE. Standard effusion cells were used to provide the molecular beam of Ga and In, as well as Ge and Si, while nitrogen was supplied by a radio-frequency Veeco plasma source. To ensure the same growth temperature, before each growth, the desorption time of the reference In and Ga fluxes was confirmed by laser reflectometry [27]. In and Ga desorption was used to confirm InGaN and GaN growth temperatures were 650 °C and 730 °C, respectively. For all InGaN layers, growth was conducted using a constant indium flux of 1 µm/h. To monitor and limit the amount of excess metal during each growth process, all effusion cell shutters were closed periodically to observe metal desorption. During each growth interrupt, a gallium or indium adlayer was fully desorbed without changing the growth temperature, with no effusion cells opened. The Ga desorption rate, as a function of the real substrate temperature, was calibrated separately on a 2-inch GaN/Al_2_O_3_ reference wafer using a Bandit pyrometer (k-Space Associates, Inc. Dexter, MI, USA). Commercially available *n*-type bulk GaN substrates with threading dislocation density (TDD) in the order of $1\;\times \;10^{7}$ cm^−2^ and a miscut angle of 0.7 degrees were used for all growths.

Doping profile measurements of stacks of GaN and InGaN layers grown at different conditions were conducted at EAG Laboratories (East Windsor, NJ, USA) using secondary ion mass spectrometry (SIMS). For clarity, all presented Ge and Si concentrations are shown assuming the same SIMS sputtering rate for GaN and InGaN, which seems to be a good approximation in the case of fully strained, low-(<10% In)-composition layers. Post-growth, surface morphology was investigated using atomic force microscopy (AFM) (Veeco, Plainview, NY, USA).

Morphology and chemical composition analysis of GaN:Ge layers were carried out using a field-emission scanning electron microscope (SEM) Hitachi SU-70 (Hitachi, Tokyo, Japan) equipped with a Thermo Fisher Scientific Ultra Dry energy-dispersive X-ray spectroscopy (EDX) system. In EDX studies, an acceleration voltage of 15 kV was applied, and Ga, Ge and N K line series were analyzed for mapping and quantitative analysis.

## 3. Results

We started the investigation by comparing doping profiles for GaN and InGaN layers grown using relatively low germanium fluxes obtained using a Ge effusion cell set to temperatures between 800 °C and 850 °C. For GaN:Ge grown at substrate temperature of 730 °C under Ga-rich conditions, this resulted in a Ge concentration between $3\;\times \;10^{16}\;\mathrm{atoms}/{\mathrm{cm}}^{3}$ and $1\;\times \;10^{17}\;\mathrm{atoms}/{\mathrm{cm}}^{3}$. Growth was periodically interrupted for 30–60 s to allow for metal (Ga or In) excess desorption, as depicted in Figure 1 by dashed lines with blue diamonds. To resolve the abruptness of the doping profile, GaN layers doped with different germanium fluxes were separated with layers that were intentionally grown with the germanium cell closed (Ge was not supplied). In spite of the fact that the Ge shutter was closed, the Ge concentration in GaN, depicted on the right-hand side of Figure 1, showed no abrupt change in Ge concentration, and the obtained profile was rather dissolved. Interfaces between GaN:Ge and undoped GaN layers, which are fingerprints of the “memory effect” that is manifested as a relatively high unintentional doping level persisting after closing of the dopant source, were very hard to identify. This could indicate that Ge can be either dissolved in a metallic Ga adlayer or can float on top of it. At the depth of 350 nm (Figure 1), the growth temperature was lowered to 650 °C, to ensure the optimal temperature for the use of indium as a surfactant, and InGaN growth started. First, a 30 nm-thick layer (depth 320–350 nm) was grown with no Ge supplied. Then, two InGaN:Ge layers utilizing different Ge cell temperatures were grown. Between doped layers, we used the undoped layer to resolve the abruptness of the doping profile.

Contrary to GaN growth, for InGaN the Ge doping profile presented in Figure 1 is sharp and, as desired, all Ge atoms were incorporated into the crystal rather than remaining dissolved in the metallic adlayer. Moreover, as can be deduced from the sharp peak in Ge concentration near the depth of 350 nm in Figure 1, all Ge atoms accumulated on the surface during GaN:Ge growth (depths 780–530 nm), causing the substantial background doping at depths between 530 and 350 nm, and became incorporated immediately as the growth of InGaN layer started. 

To explicitly present the difference in Ge incorporation into InGaN and GaN layers, a structure comprising three layers, A, B and C, was grown at the same temperature (650 °C), constant growth rate (0.85 µm/h), as defined by RF plasma parameters, and Ge flux, as defined by the constant Ge cell temperature (1150 °C), but with varying surfactants (In or Ga) and Ga/N flux ratios. Layers A, B and C were InGaN with In as the surfactant, GaN with In as the surfactant, and GaN with Ga as the surfactant, respectively. The type of metallic adlayer and Ga/N ratio for each of the doped and capping layers are listed in Table 1. Resulting Ge and In compositions are presented on the left-hand side of Figure 2. In each case, the doped layer continuously grew, with an interruption at the end, followed by the growth of an intentionally undoped layer. Schematic images of the surface and metal adlayers are presented on the right-hand side of Figure 2 labeled corresponding to markers included in the SIMS profile.

For layer “A”, indium was used as a surfactant, and a gallium flux lower than the nitrogen flux was used to enable indium incorporation into the layer [28]. At such conditions, all Ga and Ge atoms were incorporated during growth (as schematically depicted in (A1)), so after the growth interruption (A2), when undoped growth resumed, Ge concentration dropped sharply and indium composition stayed constant, as depicted in layer “A” in Figure 2. The noticeably smaller indium content right next to the growth interruption (to about 5% or so) was most probably caused by a gallium cell transient. 

For layer “B”, indium was also used as a surfactant, but the gallium flux was slightly higher than the nitrogen flux, resulting in negligible indium incorporation and accumulation of gallium on the surface [28]. This led to the situation where not all Ga and Ge atoms were incorporated during growth (B1), and after the interruption, only excess indium could be desorbed (B2). This point is indicated by a dashed line and diamond. When growth was resumed, using similar fluxes as for section “A”, first, accumulated Ga became incorporated, and soon afterwards indium composition jumped from ≈ 0 to about 10%. When excess gallium became incorporated, all Ge atoms that were not built into the crystal during the growth of the doped part became immediately incorporated, causing a spike in Ge concentration near (B2). The total (integrated) amount of Ge detected by SIMS in “A” and “B” was the same, within a 2% discrepancy, which points to negligible Ge desorption at the used growth temperature.

For layer “C”, no indium was introduced during growth (C1). Instead, the gallium flux was set to be about 10% higher than the nitrogen flux to ensure gallium accumulation, leading to the formation of the gallium wetting layer. This resulted in significantly lower germanium incorporation throughout the doped layer. After the intentionally doped part, 30 s interruptions were introduced every 25 nm of further growth of intentionally undoped GaN. As shown in Figure 2, in layer “C”, each interface could be clearly resolved as it was accompanied by a steep change in germanium concentration. Even after four periods of 25 nm-thick GaN layers, with no extra Ge introduced, the germanium dissolved in gallium present on the surface (C2) behaved as a dopant reservoir, provoking Ge doping at the level of $2\;\times \;10^{19}\;\mathrm{atoms}/{\mathrm{cm}}^{3}$.

To further analyze what is happening with germanium atoms that do not incorporate into GaN crystal, but stay at the surface, we investigated the surface of the 500 nm-thick, continuously doped GaN:Ge layer using an SEM setup equipped with EDX. The layer was grown using the same Ge flux as for layers presented in Figure 2. The surface after growth was decorated with crystallites, which were also visible under an optical microscope. A typical SEM image is presented in Figure 3a. The crystallites exhibited clear crystallographic 3-fold symmetry. The elemental distribution map (Figure 3b) confirmed that these crystallites are extremely Ge-rich. EDX spectra for crystallite and away from crystallite are compared in Figure 3c,d, respectively. Since the layer was grown at about 730 °C and under excess gallium, which was desorbed after growth, we concluded that these crystallites were most likely pure germanium crystals that were grown from solution in the liquid gallium.

Finally, to compare the results presented above for germanium with silicon, which is most commonly used for *n*-type dopants, we grew a SIMS stack where both dopants were used. This allowed for the direct presentation of different behaviors for Si and Ge dopants, in the same sample and under equivalent conditions. Layers utilizing different dopants were alternated as depicted in the SIMS profile presented in Figure 4. To prove that the observed peculiarity for Ge-doped layers occurred in a wide spectrum of growth conditions, first, two InGaN layers (depth 600–470 nm) were grown at a growth rate of 0.85 µm/h and later (between 470 and 340 nm) at 0.35 µm/h. The growth rate was controlled using the nitrogen flux. The temperature of Si and Ge effusion cells was changed between layers to match the values indicated in Figure 4. The highest Ge concentration was obtained for Ge effusion cell temperature of 1150 °C and a growth rate of 0.35 µm/h used in InGaN, which resulted in a Ge concentration of $6.5\;\times \;10^{20}\;\mathrm{atoms}/{\mathrm{cm}}^{3}$. No difference in sharpness of the profiles between Si and Ge in InGaN was obtained.

Near the depth of 340 nm, the growth temperature was increased to allow for desorption of the gallium excess. Further growth was conducted in Ga-rich conditions without introducing In. Contradictory to the Ge peak observed for the transition from GaN to InGaN growth present near 350 nm in Figure 1, for the transition from InGaN to GaN, high doping background in intentionally undoped GaN was obtained (depth 340–225 nm in Figure 4). The doping background decreased from $2\;\times \;10^{19}\;\mathrm{atoms}/{\mathrm{cm}}^{3}$ (near 340 nm) to $1\;\times \;10^{18}\;\mathrm{atoms}/{\mathrm{cm}}^{3}$ (at 225 nm), which might indicate the presence of a finite Ge reservoir at the surface. Between the depths 225 and 340 nm, for reference with Ge doping, intentional Si was introduced during growth. In the case of Si, the doping profiles shown in Figure 4 were very sharp. No difference between growth under the Ga and In excess for Si doping was found. Between the depths of 110–225 nm, Ge was intentionally introduced during growth in the same conditions as those used in the Si-doped layer proceeding it.

Concerning the electrical activity of introduced dopant, using the indium wetting layer, we obtained low-indium-content InGaN:Ge layer with one of the highest reported electron concentrations (n_Hall_) in *n*-type doped nitrides reaching $7.9\;\times \;10^{20}\;{\mathrm{cm}}^{-3}$ as obtained by Hall measurement [29]. n_Hall_ for InGaN:Ge, together with InGaN:Si and GaN:Si samples for reference, as a function of dopant concentrations estimated by SIMS, are presented in Figure 5. In the wide doping range of interest, from $1\;\times \;10^{19}$ to $1\;\times \;10^{21}\;\mathrm{atoms}/{\mathrm{cm}}^{3}$, the concentration of the free carriers corresponded exactly to the number of the introduced dopants as estimated by SIMS. The highest concentration of germanium obtained by SIMS in InGaN:Ge samples was close to $2\;\times \;10^{21}\;\mathrm{atoms}/{\mathrm{cm}}^{3}$, which was limited by the used Ge flux. In that case, electron concentration was lower: $5\;\times \;10^{20}\;{\mathrm{cm}}^{-3}$. The presented dependence of n_Hall_ versus dopant concentration suggests that Ge incorporation occurs dominantly in donor sites up to concentrations around $8\;\times \;10^{20}\;\mathrm{atoms}/{\mathrm{cm}}^{3}.$ Differences between Ge and Si in n_Hall_ at low doping concentrations are most probably caused by lower accuracy of flux extrapolation in that range. 

The InGaN:Ge morphology was also investigated as a function of Ge doping to assess its quality. In this case, 30 nm-thick InGaN:Ge layers with no cap were grown on 100 nm GaN buffer using different Ge cell temperatures. The resulting morphologies are presented in Figure 6. The root mean square (RMS) value increased with increasing germanium concentration, which, at least to some extent, can be engineered by further optimization of the main heater correction needed due to the change in surface temperature caused by a significant change in absorption [16]. Atomic steps could be clearly resolved up to $1\;\times \;10^{20}\;\mathrm{atoms}/{\mathrm{cm}}^{3}$. At this level, atomic steps started to be wavy. Adding even more Ge resulted in higher roughness that might have been due to the postulated effect of Ge atoms on the mobility of Ga atoms [30].

## 4. Discussion

Previous reports on the limitations of Ge doping level during GaN growth investigated the restrictions imposed by the crystal itself. This study showed that the state of the surface during the growth, especially the type of surfactant metal, can be more discriminative towards dopant incorporation than the crystal itself. Presented experimental results indicate pronounced difficulties in obtaining high Ge concentrations in GaN grown under Ga-rich conditions, typical for plasma-assisted MBE. It is important to note that the observed “memory effect”, caused by Ge atoms accumulating or dissolving in the Ga adlayer, was also present for relatively low doping concentrations, as presented in Figure 1. The “memory effect”, if not properly managed, will have even more severe repercussions on the growth of heavily doped heterostructures, such as tunnel junctions, where precise control of doping profiles and interfaces determines the device performance. For thick, heavily doped GaN:Ge layers, it also leads to Ge precipitates, as reported before by others [25] and observed also for our crystals.

In the literature, the presence of a limiting mechanism for Ge incorporation into GaN is supported by reports of the use of Ge as a catalyst for the growth of GaN nanowires (NWs) [31]. In that study, Ge deposited onto Si surface prior to growth formed droplets that catalyze NW growth and lead to relatively small (~10^19^ $\mathrm{atoms}/{\mathrm{cm}}^{3}$) germanium contamination [32].

The doping profiles and incorporation of Ge atoms into nitride layers completely changed when Ga is replaced by an In adlayer. In the case indium was used as a surfactant, all impinging Ge was incorporated into the crystal rather than staying at the surface. This change was strictly associated with the element used as a metallic surfactant. It can be concluded from the fact that growth at the same temperature for the Ga surfactant led to a pronounced “memory effect” in the composition profile in contrast to the In-surfactant that facilitated abrupt doping interfaces. Importantly, such an effect was not expected judging from solubility diagrams for Ge in Ga and In. Both elements offer high solubility, with a slightly higher value for Ga (55%) than for In (45%). 

Furthermore, no difference between doping profiles for the Ga and In adlayer for the Si-doped III-nitride layers was obtained. No “memory effect” or dissolved interfaces present for Ge-doped layers were observed for this Si profile.

The most plausible cause for distinct incorporation mechanisms for Ge and Si atoms for different kinds of adlayers is the alteration in diffusivity of the dopants. This, in turn, might be connected with the relative size of the atoms. Since the Ge atom is slightly larger than the Ga atom, the dopant diffusing through an organized Ga adlayer is blocked. For the much smaller Si atom, the same adlayer might not present a significant diffusion barrier. The In adlayer, on the other hand, consists of atoms larger than Ge, which could explain why Ge, in this case, can more easily diffuse through it, resulting in the sharp doping profiles and the lack of “memory effect” observed in that case.

## 5. Conclusions

This study analyzed the impact of surfactants—gallium and indium—on dopant incorporation in GaN and InGaN layers grown by PAMBE. Layers grown using the Ga adlayer suffered from a dissolved doping profile in the structure regardless of the doping level. Additionally, Ge precipitates forming on the surface were found. Contrary to this, InGaN and GaN grown using an In adlayer were characterized by sharp doping profiles and led to higher Ge concentrations. The more commonly used Si dopant did not present such an anomaly, and for both the Ga and In adlayer, the doping profile was sharp.

The presented findings show that the most efficient way of obtaining heavily Ge-doped layers with a controlled profile is to use indium as a surfactant. The sharpness of the doping profile is the most important factor for heterostructures such as tunnel junctions, p-n diodes or vertical transistors, where residual doping is detrimental to the final device performance. For specific applications, where a high crystalline quality at a high doping level is essential, Ge offers a significant advantage over Si.

The use of indium as a surfactant (instead of gallium) limits the accessible growth temperature due to the more pronounced indium desorption. For the growth of heavily doped layers, the stable metallic coverage at the surface can become more complex. Further work will focus on the stabilization of the growth temperature during the growth of heavily doped InGaN:Ge layers. Advantages offered by indium surfactant in germanium incorporation could also prove to be applicable to other alloys. For example, it could pave the way towards efficiently doped high-electron concentration InAlN.

## Figures and Tables

**Figure 1 materials-15-05929-f001:**
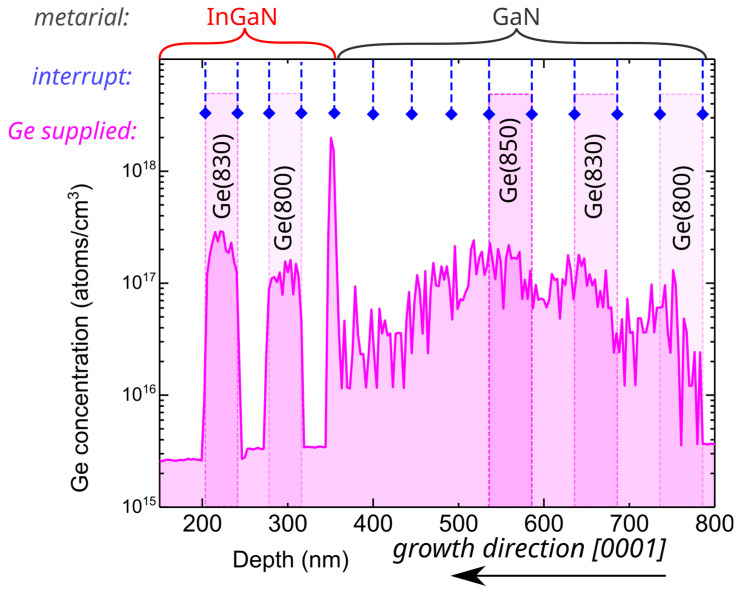
Germanium concentration measured by SIMS as a function of depth for a stack of InGaN:Ge and GaN:Ge layers separated by layers of the same composition that were intentionally undoped. InGaN and GaN parts of the crystal are marked above the plot. Places where the germanium effusion cell was opened are marked by additional shading with germanium cell temperature in Celsius. Places where all atomic fluxes were closed to allow for excess gallium and indium desorption are marked with dashed blue lines and a diamond symbol.

**Figure 2 materials-15-05929-f002:**
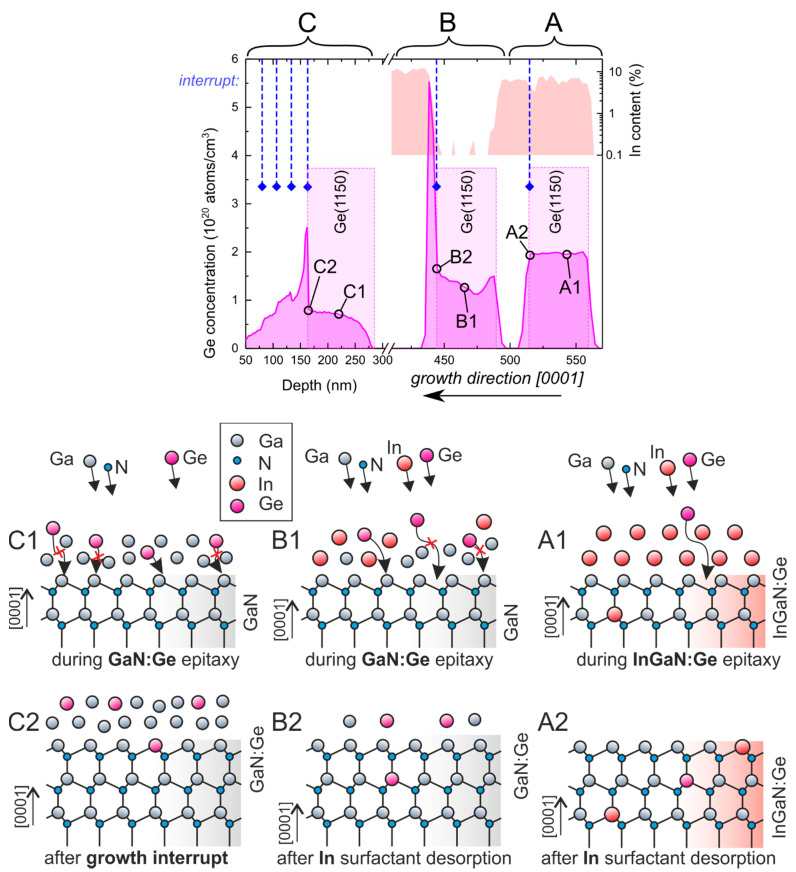
Germanium concentration measured by SIMS (linear scale) as a function of depth for a stack of InGaN:Ge and GaN:Ge layers separated by intentionally undoped layers. Indium content as a function of depth is shown on the right-hand side axis. Places where germanium effusion cell was opened are marked by additional shading with germanium cell temperature in Celsius (which was kept constant in this experiment). Places where all atomic fluxes were closed to allow for excess gallium and indium desorption are marked with dashed blue lines and a diamond symbol. Parts (**A**–**C**) show distinct parts of the doping profile obtained on a single sample. Cartoons on the bottom part schematically depict state of the surface at points A1, A2, B1, B2, C1 and C2 as marked in the doping profile.

**Figure 3 materials-15-05929-f003:**
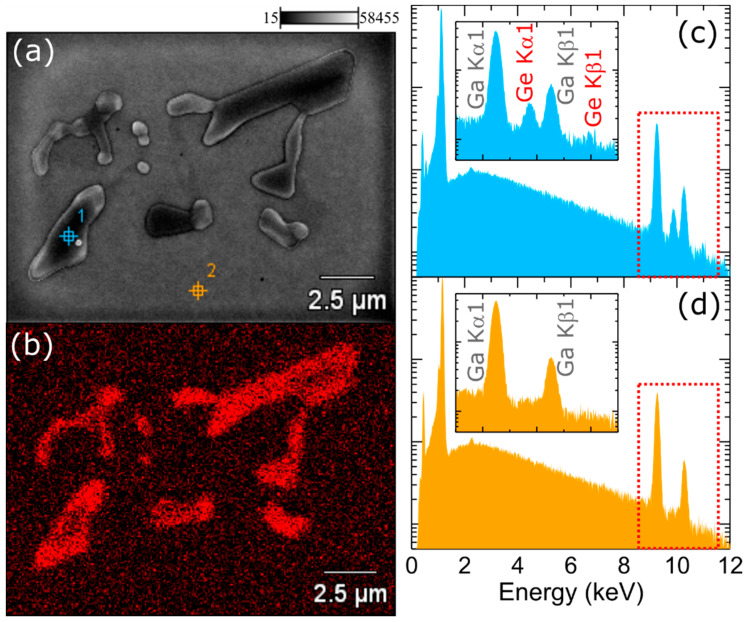
(**a**) Plane-view SEM image of the surface of 500 nm-thick GaN:Ge layer and (**b**) a corresponding energy-dispersive X-ray spectroscopy (EDX) mapping of Ge Kα1 transition. EDX spectra obtained within a crystallite (spot 1 in (**a**)) and away from the crystallite (spot 2 in (**a**)) are presented in (**c**,**d**), respectively. Insets in (**c**,**d**) show the zoomed part of the spectra where the presence of Ge can be identified.

**Figure 4 materials-15-05929-f004:**
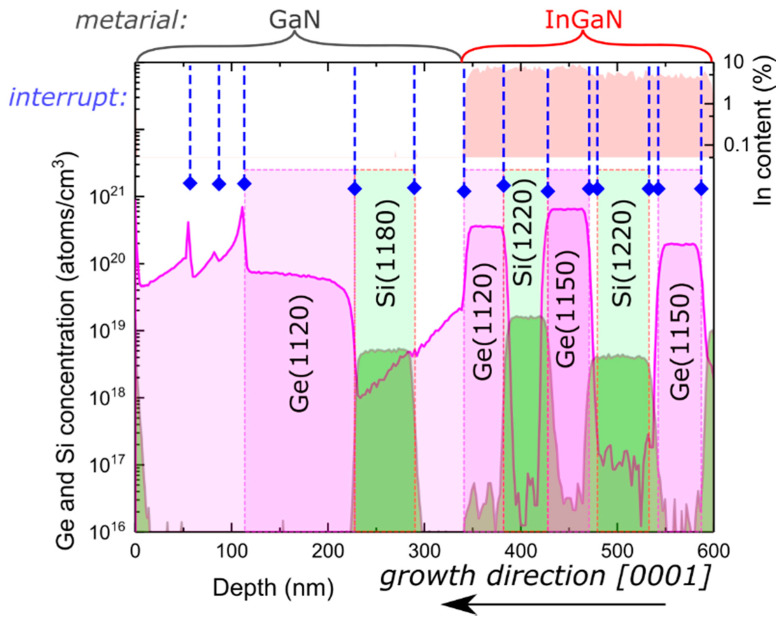
Germanium concentration measured by SIMS as a function of depth for a stack of InGaN:Ge and GaN:Ge layers separated by layers of the same composition that were intentionally undoped or doped with silicon. InGaN and GaN parts of the crystal are marked above the plot. Indium content as a function of depth is shown on the right-hand side axis. Places where germanium effusion cell was opened are marked by additional shading with germanium cell temperature in Celsius (which was kept constant in this experiment). Places where all atomic fluxes were closed to allow for excess gallium and indium desorption are marked with dashed blue lines and a diamond symbol.

**Figure 5 materials-15-05929-f005:**
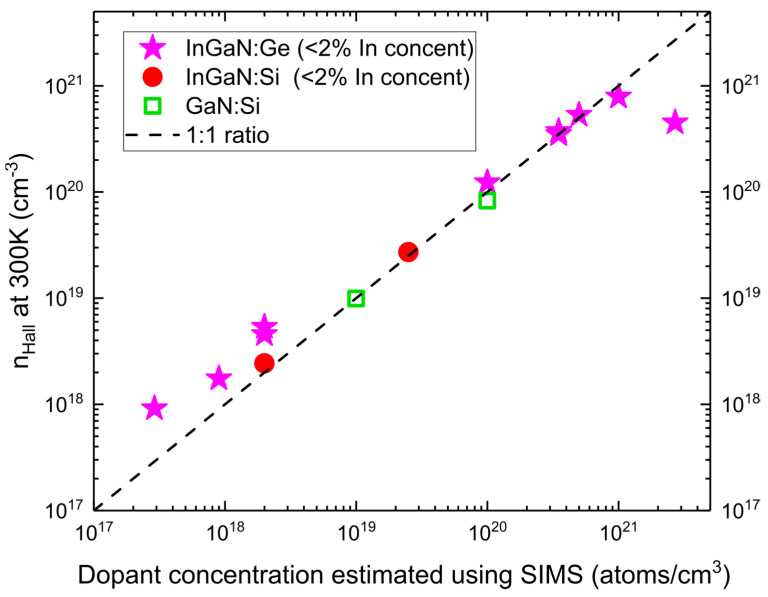
Electron concentration extracted from Hall measurement (n_Hall_) conducted at room temperature for about 300 nm-thick layers grown on semi-insulating GaN/Al_2_O_3_ templates as a function of dopant concentration estimated using secondary ion mass spectroscopy (SIMS) measurements performed on separate calibration samples. Results of Hall concentration presented in [29]. Data obtained for InGaN:Ge, InGaN:Si and GaN:Si are presented using blue stars, red circles and magenta squares, respectively. The 1 to 1 n_Hall_ electron concentration to dopant concentration is marked by a dashed line.

**Figure 6 materials-15-05929-f006:**
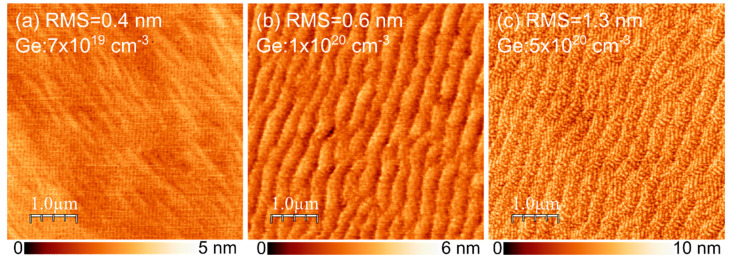
InGaN:Ge surface obtained by AFM for 30 nm-thick layers with Ge concentration tuned between $7\;\times \;10^{19}\;\mathrm{atoms}/{\mathrm{cm}}^{3}$ and $5\;\times \;10^{20}\;\mathrm{atoms}/{\mathrm{cm}}^{3}$. Corresponding root mean square (RMS) values and concentrations are indicated on each image.

**Table 1 materials-15-05929-t001:** Type of surfactant (In, Ga or mixture of the two) and Ga/N ratio used during growth of doped and capping layers for regions A, B and C marked in Figure 2. In flux and nitrogen flux (determining the growth rate) were kept constant for all three layers at 1 µm/h and 0.85 mm/h, respectively.

Layer	Doped Layer		Capping Layer	
	Surfactant	Ga/N Ratio	Surfactant	Ga/N Ratio
A	In	0.92	In	0.92
B	In+Ga	1.03	In	0.9
C	Ga	1.1	Ga	1.1

## Data Availability

The data presented in this study are available on request from the corresponding author.

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
