# Peer review of "Role of Metallic Adlayer in Limiting Ge Incorporation into GaN"

_materials, 2022, doi:10.3390/ma15175929_

Round 1

Reviewer 1 Report

The paper of Turski at al. dealing with Ge and Si dopant behavior during GaN and InGaN growth is very interesting and provides useful knowledge about incorporation of these dopants into (In)GaN. The paper is well organized and easy to follow, however in my opinion there are some issues should be addressed before the publication in Materials. Bellow I listed my remarks and questions:

- first of all, in my opinion in the paper where doping is investigated the authors should enhance SIMS analysis with ECV measurements. It will be very important to show the ionization ratio of Si and Ge dopant especially for the last grown sample where Si and Ge doped layers are alternated and additionally grown under different doping conditions (dopant flux, dopant source temperature, III/N ratio). Moreover what about Ge or Si atoms incorporated on N site, did authors somehow estimate this amounts?

- Is there any explanation why Ge doping memory effect is observable in GaN and is invisible in InGaN?

- please place the figures just after first mention in the main text

- I would enlarge the figures A1 to C2 of Figure 2

- line 28 missing subscript in N2

- please uniform the notation of dopant atom concentration: line 35 vs 60 for instance

-

Reviewer 2 Report

·       Add more new references.

·       Show the novelty of the paper compared to the literature, however there is still much work on this topic.

·       Why you choose these materials?

·       In the Introduction section, the last paragraph should contain the scientific contribution and scientific hypotheses of your research. Complete, further elaborate the scientific contribution and scientific hypotheses of your research. Be explicit. In addition to the goal of the research (which was written), the novelty in the context of the scientific contribution should be pointed out. Scientific contributions should be written based on the shortcomings of previous research in the literature. In this way, the authors will better emphasize novelty and scientific soundness.

·       Analyze and discuss possibilities of practical application.

·       Complete the conclusions with the limitations of the proposed methodology. Also write future research.

·       Generally, the quality of the writing could be improved.

Round 2

Reviewer 1 Report

I accept authors response on my remark concerning ECV measurements. I appreciate that authors include new Figure 5 and corelates the SIMS and Hall measurement findings. However please enhance the quality of Figure 5.